# Enhancing EFL Learners’ Self-Efficacy Beliefs of Learning English with Emoji Feedbacks in CALL: Why and How

**DOI:** 10.3390/bs12070227

**Published:** 2022-07-12

**Authors:** Yen-Jung Chen, Liwei Hsu

**Affiliations:** 1Institute of Education, National Sun Yat-sen University, No. 70, Lianhai Rd., Gushan Dist., Kaohsiung City 804201, Taiwan; jungjunger@gmail.com; 2Graduate Institute of Hospitality Management, National Kaohsiung University of Hospitality and Tourism, No. 1, Song-he Rd., Hsiao-Kang District, Kaohsiung City 812301, Taiwan

**Keywords:** self-efficacy beliefs, feedback, emojis, subliminal feedback, online learning, EFL learning

## Abstract

Encouraging feedback positively affects learners’ self-efficacy; in language learning, self-efficacy predicts language learner performance and behavior. Our research involved three studies to expand knowledge about why and how we can enhance English as a Foreign Language (EFL) learners’ self-efficacy beliefs in online settings. In Study 1, based on an online survey with 310 participants, we ascertained the extent to which EFL learners with greater self-efficacy tend to challenge themselves by learning content that requires a proficiency level that is higher than their current proficiency. In Study 2, we recruited 120 EFL learners; the results indicate that positive feedback via emojis embedded in online courses could significantly boost EFL learners’ self-efficacy beliefs about learning English. Study 3 involved 35 participants and extended the understanding provided by the first two studies, showing that EFL learners not only like to use emojis for computer-mediated communication (CMC), but also prefer to receive them as feedback. This research adds to knowledge on “why” and “how” we can enhance EFL learners’ self-efficacy beliefs in online contexts. We systematically provide empirical evidence regarding the aforementioned issues and demonstrate that positive feedback through emojis has great potential to enhance EFL learners’ self-efficacy, even when such feedback is subliminal.

## 1. Introduction

Feedback is critical for students’ successful learning [1], as providing effective feedback can improve their self-efficacy beliefs and boost their learning effectiveness [2]. Bandura (1986) stressed that many students experience learning difficulties in school not because they are incapable of performing successfully, but rather because they are incapable of believing that they can perform successfully [3]. In other words, many students do not have enough self-efficacy in learning. Self-efficacy belief refers to the awareness of one’s ability to perform specific tasks with satisfactory outcomes [4]. When faced with academic challenges, students with high self-efficacy tend to challenge themselves by learning target content above their current level, whereas students with low self-efficacy tend to avoid such a challenge [5]. Further, students’ self-efficacy beliefs are positively linked to their motivation to learn [6,7,8,9]. This association is also sustained in English as a foreign language (EFL) learning [8,10]. For the present study, we aimed to expand knowledge about why and how we can enhance EFL learners’ self-efficacy beliefs in computer assisted language learning (CALL) settings by providing positive feedback through emojis that is consciously and/or subconsciously perceived by the student.

Encouraging feedback has a positive effect on students’ self-efficacy [2], and in the field of language learning, self-efficacy is an important predictor of learner performance and behavior [11,12]. However, most EFL learners do not seem to have a high self-efficacy belief about learning English [13,14,15]. Under such circumstances, within either the traditional face-to-face context or online settings, teachers should understand how to effectively give students feedback without compromising their self-efficacy beliefs [16].

One advantage of online learning is that students can receive timely feedback [17]. Moreover, it is common practice to use emojis (also called emoticons) to replace facial expressions to either express a feeling or facilitate socialization in computer-mediated communication (CMC) [18,19,20,21], as emojis are textual paralanguages [22]. In online learning contexts, the use of electronic feedback to mitigate face-induced threats (through face-threat mitigation, or FTM) is a possible solution for maintaining a positive relationship between teachers and students [23]. In conventional teaching environments, feedback has traditionally been designed to be clearly and consciously perceived by the student, but whether subliminal feedback provided in online settings can have similar effects is still unknown. Subliminal information can be well perceived by individuals in an unconscious state [24,25,26], and the use of subliminal information in advertising and marketing has a long history. Per suggestions of previous studies, Ionescu (2016) proposed that stimuli that last 30–50 milliseconds can be considered subliminal stimuli [27]. Carr (2011) and Weber (2016) further set up the threshold value of subliminal stimuli at 30 milliseconds [28]. In the current study, subliminal feedback refers to the feedback that appears for less than 30 milliseconds (or. 0.03 s) and is unconsciously attained by the receivers. When such feedback is presented with emojis and could be unconsciously perceived by the students, it serves as subliminal feedback through emojis. Conversely, when conscious feedback in emojis were captured consciously by the students, these accounted for the supraliminal feedback. Additionally, the subconscious mind influences human cognition, indicating that subliminal stimuli can evoke specific knowledge and beliefs, especially when these stimuli delineate the current needs or desires of a specific object [25,29,30,31]. Williams et al. (2004) [32], as well as Winkielman and Berridge (2004) [33], provided empirical evidence that subliminal messages have particular effects on the recipient’s behavior and brain activity. Arslan et al. (2017) further suggested that subliminal messages or feedback can be used in various fields, including education, to expand the applicability of subliminal messages/feedback in educational praxis [24].

As put forth by Bargh and Morsella (2008) “the unconscious mind is a pervasive, powerful influence over such higher mental processes” (p. 73) [34]. Hamad (2007) also pointed out that one’s unconscious mind is able to facilitate one’s creativity and accomplish what the conscious one could not [35]. Since the unconscious mind is more powerful than the conscious one [36], we focused on needs and assumptions with specific arguments; that is, if encouraging feedback is delivered to EFL learners as subconscious information, their self-efficacy beliefs about EFL learning will be significantly enhanced. Given the positive effect of encouraging feedback on students’ self-efficacy [2], establishing an effective subconscious feedback mechanism in language learning is a vital part of the educational environment. As such, we conducted three studies to shed light on the relationship between critical elements of language learning: (1) EFL students’ self-efficacy beliefs, and (2) their inclination to challenge themselves in EFL learning. Moreover, we investigated whether positive feedback via emojis would improve EFL students’ self-efficacy beliefs. Lastly, we explored whether subliminal feedback would effectively be able to enhance and promote EFL learners’ self-efficacy beliefs. To the best of our knowledge, this study is the first empirical research to provide evidence of how using emojis as feedback influences EFL learners’ self-efficacy beliefs. To address this issue, we reviewed pertinent literature to formulate hypotheses, and we designed three studies to test them. The first study involved a survey to confirm the association of EFL learners’ self-efficacy beliefs about EFL learning and their intention to learn something beyond their current level. The second and third studies entailed experiments to test how positive feedback through emojis (both supraliminal and subliminal) enhances EFL learners’ self-efficacy beliefs.

## 2. Prior Studies and Development of the Hypotheses

### 2.1. Self-Efficacy Beliefs and EFL Learning

The term self-efficacy was coined by Bandura (1986), defined as one’s belief about one’s capability to accomplish a designed task with an expected outcome [3]. In the field of language learning or second language acquisition (SLA), an individual’s self-efficacy beliefs can predict their performance [12,37,38,39,40,41,42]. Furthermore, self-efficacy beliefs can positively influence a student to put effort into learning [10], and students’ self-efficacy beliefs are domain specific [43]. Shin (2018) and Wang et al. (2013) claimed that EFL learners with greater self-efficacy beliefs will be more willing to accept challenges, which in this study refer to the vocabulary lists that were beyond the participants’ current vocabulary volume or were above their current proficiency level [8,44]. However, no empirical data have been presented in the research to support this assertion. We attempted to fill this gap with empirical evidence and hence formulated the first hypothesis:

**H1.** 
*EFL learners with greater self-efficacy beliefs tend to challenge themselves by learning content that requires a proficiency level higher than their current proficiency level.*


### 2.2. Positive Emoji Feedbacks and EFL Learners’ Self-Efficacy

The term feedback, in the academic context, was coined by Hattie and Timperley (2007) [45], defined as information given by teachers or peers based on their evaluation of one’s performance of a specific task. Feedback is a key element of students’ successful learning [46]. Gan et al. (2021) demonstrated the importance of feedback for EFL learners’ English writing ability [47]. Using computer-based feedback, Sherafati and Mahmoudi Largani (2022) demonstrated that EFL learners’ self-efficacy beliefs about writing English can be significantly improved [48]. For example, regarding previous research on providing students with feedback through text, Moffitt et al. (2021) conducted a study to explore the use of emojis for giving students feedback [49]. Moreover, positive feedback in the form of emojis is likely to be used in online text-based communication [50,51]; therefore, emojis can be employed to facilitate communication in online learning [52]. Jin (2018) reported that Chinese language learners are keen to talk with their peers about interesting topics by using emojis [53]; this finding is supported by Luo and Gao (2022) [54]. Chen et al. (2021) also found that Chinese EFL learners are eager to use emojis as a part of their socialization strategies in online language learning [55]. The participants in their study utilized positive emojis such as “thumb up” or “smiling face” to express support for their peers. Al-Garaady and Mahyoob (2021) revealed that the EFL learner participants of their study liked to use emojis for communication [56], but such an application is likely to have negative effects [57,58]; specifically, emojis could negatively affect their writing skills as their standard writing might be sacrificed because of these convenient and emergent communication tools. Liu et al. (2021) examined the use of peer feedback in face-to-face and online EFL writing courses; the Chinese EFL learners in their study indicated that emojis can be beneficial in cultivating positive emotions and building rapport among peers [59].

Although a growing body of research has addressed the potential use of emojis for online language learning, as well as how one’s self-efficacy beliefs influence one’s reaction to feedback [60], studies on the applications of emojis to enhance EFL learners’ self-efficacy beliefs remain limited. As such, we formulated the second hypothesis:

**H2.** 
*Positive feedback presented supraliminally/consciously in the form of emojis will significantly enhance EFL learners’ self-efficacy beliefs.*


### 2.3. Supraliminal or Subliminal Positive Feedback

Extensive studies have been performed on how subliminal messages influence decisions and behavior [61]. Such unconsciously noticed messages affect emotions, semantic performance [62], and cognitive performance [63]. Subliminal cues or feedback cannot be captured by an individual’s explicit learning system; hence, the implicit learning system will be activated to apprehend subliminal cues [64]. Ham et al. (2009) designed an experiment to understand the impact of subliminal feedback on participants’ attitudes toward energy consumption of household appliances [65]; the participants who received subliminal feedback outperformed those who did not. In another study, Ruijten et al. (2011) found that subliminal feedback boosted participants’ performance in terms of energy-related choices [66]. However, previous studies emphasize the presence of goal-striving to optimize subliminal feedback, and it remains unclear whether positive subliminal feedback in the form of emojis would enhance EFL learners’ self-efficacy beliefs. As such, we formulated the third hypothesis:

**H3.** 
*Positive feedback in the form of emojis presented subliminally and supraliminally will significantly affect EFL learners’ self-efficacy beliefs.*


## 3. Materials and Methods

We carried out three studies to address our hypotheses. For the first two studies, we focused on the relationship between EFL learners’ self-efficacy beliefs and their willingness to challenge themselves while learning, as well as whether using positive feedback in the form of emojis in an online environment would affect their self-efficacy beliefs. The results of the first two studies provided answers for H1 and H2. For the third study, we designed and administered an experiment to understand whether positive feedback in the form of emojis presented subliminally would have a similar impact on EFL learners’ self-efficacy beliefs. We performed statistical analyses for all three studies using SPSS 25 (IBM Corp. Released 2017. IBM SPSS Statistics for Windows, Version 25.0. Armonk, NY, USA: IBM Corp).

### 3.1. Study 1

In Study 1, we aimed to measure whether EFL learners’ self-efficacy beliefs significantly affects their willingness to learn challenging English vocabulary. To extend the scope of the survey, we administered it online to scrutinize H1.

#### 3.1.1. Participants and Design

We conducted Study 1 online. A purposeful sampling technique was adopted to recruit the participants via social media platforms such as Facebook and Line groups. To ensure the representativeness of the participants, their TOEIC (Test of English for International Communication) score had to be around 560 (one question asks the participant’s TOEIC score; individual participants who have not taken the TOEIC yet or whose TOEIC was far below 560 were removed from further analysis), since this score was the average of Taiwanese EFL learners in the year 2020 (ETS, 2020); 310 (n = 310) EFL Taiwanese learners took part in the survey. Their demographic traits included gender (females = 169, 54.5%; males = 141, 45.5%) and age (M = 21.3, SD = 2.6). Once they agreed to participate, they could start the survey. Since current research stresses the importance of vocabulary in language learning [67,68], we narrowed the focus of Study 1 to EFL learners’ self-efficacy beliefs about vocabulary learning and included questionnaire items on self-efficacy related to reading in the survey. At the end of the survey, the participants were asked which groups of vocabulary they were keen to learn (Group 1 = easy, e.g., allergy, infection, physician, politics, surgery; Group 2 = challenging, e.g., acetaminophen, melatonin, electroencephalogram); therefore, the dependent variable of Study 1 was categorical. The levels of difficulty pertaining to vocabulary lists were based on the participants’ level of proficiency. Challenging words were those selected from “Word Smart for the GRE” and these vocabulary lists were difficult for the EFL learners [69], whereas easy words were those that were used in daily conversation.

#### 3.1.2. Measurement

For Study 1, we employed the Questionnaire of English Self-Efficacy (QESE) developed by Wang et al. (2013) [8]. The QESE has been validated with satisfactory results [70]. We adapted items on EFL learners’ self-efficacy beliefs about reading (e.g., “Can you finish your English reading homework independently?”). All of the items are rated on a 5-point Likert scale (1 = strongly disagree, 5 = strongly agree); the reliability of QESE is Cronbach’s α = 0.80, and the information of a single-factor confirmatory factor analysis (CFA) for the reading section is reported in Table 1. The model fit indices denote that this single-factor model fits the data perfectly. After the participants finished the survey on their self-efficacy beliefs, they had to choose what kinds of vocabulary they were interested in learning.

### 3.2. Study 2

For Study 2, we aimed to examine how feedback provided via emojis affects EFL learners’ self-efficacy beliefs in CALL settings. We designed a between-subjects experiment with the independent variable being the presentation of positive feedback in the form of emojis, and the dependent variable being EFL learners’ level of self-efficacy beliefs.

#### 3.2.1. Participants and Design

We calculated the sample size of Study 2 using G*Power with an effect size d = 0.5, α error probability = 0.05, and statistical power = 0.8 for the independent *t*-test, which came up with a minimum number of 102 (51 for each group). Considering this, purposeful sampling was conducted and 120 EFL learners from universities in southern Taiwan were recruited as participants; their TOEIC score was also around 560. To prevent the influence of other stimuli such as sounds and lighting, the experiment was administered in a soundproof laboratory with consistent lighting source. When participants arrived at the experiment site and signed the consent form, we randomly assigned them to two groups (EG: experimental group = CALL with positive feedback via emojis, appearing for 3 s every five sample questions, four different positive emojis were used in Study 2 and 3 [see Figure 1]; CG: control group = CALL without such feedback via emojis [refer to Figure 2]). Each participant was invited to a soundproof laboratory and seated 45 cm away from a 22-inch computer monitor, wherein they received a 20-min CALL lesson containing a series of sentences that frequently appear on the TOEIC test. Before the onset of this experiment, the participants were asked to fill out the questionnaire as a pre-test of their self-efficacy beliefs. The results of the pre-test indicated that there were no significant differences in self-efficacy beliefs between the two groups (t(118)= −0.48, *p* > 0.05).

#### 3.2.2. Measurement

For Study 2, to measure participants’ self-efficacy beliefs, we used the QESE mentioned earlier. We focused on EFL learners’ self-efficacy beliefs about their reading and listening comprehension; thus, we adapted 12 questions for this study. Among the 12 questions, the 6 about self-efficacy regarding reading comprehension were the same as those asked in Study 1 with some modifications to fit in this research context, whereas the other 6 questions on listening comprehension were from the QESE listening portion (sample question: “Can you understand the conversations in the previous tasks?”). The reliability test for these questions revealed Cronbach’s α = 0.89. Table 2 displays the details of a single-factor CFA of the QESE listening portion. Table 2 also confirms that this single-factor model fits the data perfectly.

### 3.3. Study 3

To better understand whether positive feedback via emojis presented subliminally can also be effective in terms of enhancing EFL learners’ self-efficacy beliefs, Study 3 involved a repeated measures within-subjects (each participant was exposed to three different designs of CALL lessons, namely, no feedback vs. 3-s feedback/supraliminal feedback vs. 0.03-s feedback/subliminal feedback) experiment to address this issue.

#### 3.3.1. Participants and Design

We calculated the sample size of Study 3 by G*Power with an effect size of f = 0.5, α error probability = 0.05, statistical power = 0.8 for repeated measures within-factor analysis of variance (ANOVA), which produced an appropriate number of 30 participants. We recruited 35 participants purposefully for Study 3 (n = 35). Once they were recruited, they were told about the nature of the study and signed informed consent forms before the experiment. All of the participants were from the same public university in southern Taiwan; again, their average TOEIC score was 560. Participants who already joined Study 2 would be excluded to avoid the maturation effect [71]. Moreover, to prevent the influence of other stimuli such as sounds and lighting, the experiment was administered in a soundproof laboratory with consistent lighting source, and the participants were seated 45 cm away from a 22-inch computer monitor. The content of the online lesson was identical to that of Study 2; the only difference lay in the inclusion of subliminal feedback as one of the treatments in the experiment. Each participant attended three 20-min lessons and received positive feedback (presented supraliminally and subliminally for 3 s and 0.03 s, respectively, for every five questions) and without positive feedback (Lesson 1: no feedback; Lesson 2: subliminal positive feedback; Lesson 3: supraliminal positive feedback). At the end of each session, the participants were asked to fill a questionnaire seeking information pertaining to their self-efficacy beliefs. Given the fact that the participants received the same CMC messages with (EG) and without (CG) positive feedback, we randomized the sequence of the three lessons and set the interval between the lessons at least 48 h to prevent the carryover effect [72]. The procedure for Study 3 is depicted in Figure 3 below.

#### 3.3.2. Measurement

We used the same measurements for participants’ self-efficacy beliefs as those employed for Study 2, and we confirmed the reliability and validity of Study 3.

## 4. Results and Discussion

### 4.1. Study 1

Among the 310 participants, 175 participants selected the easier vocabulary with a lower level of self-efficacy (M = 4.16, SD = 0.58) compared to the other 134, who chose the more challenging vocabulary with higher self-efficacy (M = 4.32, SD = 0.48). Study 1 showed a significant difference in terms of their self-efficacy beliefs about EFL learning as t(307) = −2.68, *p* = 0.008. The effect size, Cohen’s d = 0.30, confirmed that the grouping had a small to medium effect size [73]. The findings of Study 1 provide empirical support for H1 (i.e., EFL learners with greater self-efficacy tend to choose the vocabulary lists from GRE workbook, while those with low scores on self-efficacy beliefs were keen to select the vocabulary list that appeared more often in their daily life. This outcome is in line with the work of Kim et al. (2022) [10], who posited that students with greater self-efficacy beliefs tend to set up more challenging goals by taking on tasks above their present level. The results of Study 1 underscore the importance of EFL learners’ self-efficacy beliefs about their learning attitude. Thus, EFL teachers should identify effective approaches to enhance students’ self-efficacy beliefs. Liu, Du, Zhou, and Huang (2021) [59] revealed that EFL learners’ self-efficacy beliefs could be enhanced through emotional arousal, particularly positive emotion [74]. Johnson (2021) further pointed out that direct instruction can effectively improve bilingual learners’ self-efficacy beliefs [75]. To sum up, direct instruction along with positive feedback to EFL learners can significantly enhance their self-efficacy beliefs.

Furthermore, students generally are encouraged in any form of positive feedback, such as voice, text, or graphical icons. It is necessary to have a more rigorous experiment to further explore the previous finding. Furthermore, the use of emojis has been a common practice in digital interactions, which is able to convey emotional and pragmatic meaning in communication [76]. As such, we designed Study 2 to expand understanding of whether positive feedback through emojis (vs no feedback) can boost EFL students’ self-efficacy beliefs about EFL learning.

### 4.2. Study 2

The results of the independent *t*-test showed a significant difference between the two groups (t(118) = −2.58, *p* = 0.01). Such a significant difference is supported by a medium effect size (Cohen’s d = 0.46), as a medium effect size should “represent an effect likely to be visible to the naked eye of a careful observer…” and even a small effect size is “to be noticeably smaller than medium but not so small as to be trivial…”(p. 25) [77]. Further paired sample *t*-tests revealed that the control group participants’ self-efficacy did not exhibit a significant difference between the pre-test and the post-test (t(59) = −0.95, *p* > 0.05), while their experimental group counterparts did (t(59) = −2.92, *p* = 0.005). Detailed information is reported in Table 3. Such a finding supports H2, which conjectures that positive feedback via emojis embedded in online lessons will significantly enhance EFL learners’ self-efficacy beliefs about learning English. Study 2 also supports the outcomes of previous research on the constructive role that feedback plays in online learning contexts [78,79], and that feedback in the form of emojis can also boost EFL learners’ self-efficacy beliefs. However, it is still unclear whether learners need to attend to feedback in order for it to be effective. Therefore, we designed Study 3 to establish how positive feedback via emojis, presented subliminally, would influence EFL learners’ self-efficacy in English learning.

### 4.3. Study 3

We performed repeated measures ANOVA to compare the effect of positive feedback in the form of emojis on EFL learners’ self-efficacy beliefs. In Study 3, we investigated participants’ self-efficacy beliefs about listening comprehension and reading independently. The descriptive statistics are reported in the Table 4 below:

As for self-efficacy regarding listening comprehension, there was a statistically significant difference between at least two lessons (F(2, 87) = 9.36, *p* < 0.000). We noted a similarly significant outcome for reading self-efficacy (F(2, 87) = 15.90, *p* < 0.000). The effect sizes of these two parts were large ($\eta_{p}^{2}$ = 0.18 and 0.27, respectively), which confirmed the significant findings of ANOVA. As such, H3 is supported by Study 3. According to the data reported in Table 4, the participants’ self-efficacy beliefs with regard to their reading ability were higher than that with regard to their listening comprehension. The results of ANOVA are shown in Table 5. With regard to the effectiveness of subliminal and supraliminal emojis as feedback, the results of the study unveiled that both types of emoji feedback were identical, and they could significantly increase EFL learners’ self-efficacy compared to no feedback. The current study advances Ham, Midden, and Beute’s (2009) [65] finding that the effectiveness of supraliminal feedback was similar to that of subliminal feedback by explicating that emoji feedback can also work.

The results of the ANOVA were significant. A Bonferroni test for multiple comparisons revealed that the mean value of the first lesson was significantly different from those of the other two lessons (listening: *p* = 0.09 for the 2nd lesson, *p* = 0.000 for the 3rd lesson, 95% confidence interval [CI] = [−0.24, −0.03] and [−0.29, −0.08]; reading: *p* = 0.000 for the 2nd lesson, *p* = 0.000 for the 3rd session, 95% CI = [−0.44, −0.12] and [−0.51, −0.19]), while no significant difference was found between the second and third lessons. Such findings indicate that the EFL learners’ self-efficacy beliefs were significantly higher when positive feedback via emojis was included in the lessons, with no significant difference between supraliminal and subliminal feedback. This outcome echoes the work of Dao et al. (2021) [80] and González-Lloret (2015) [81], who argued that non-native speakers tend to use emojis for language learning. Furthermore, Study 3 deepens the current understanding provided by the first two studies by showing the effectiveness of positive emoji feedback that was used for EFL learners’ learning in CALL settings, regardless of whether the feedback was subliminal or supraliminal. The reasons may be more than just convenience as according to the participants from the study by Dao et al. (2021) [80], their self-efficacy beliefs in listening and reading were also enhanced. Moreover, Study 3 suggests that even feedback not consciously captured by EFL learners would be beneficial to boosting their self-efficacy beliefs. This logic aligns with the work of Hsu and Chen (2020) [25], who examined the effectiveness of subliminal stimuli in receiving messages.

## 5. General Discussion, Implications, and Conclusions

We designed three studies to examine the hypotheses, which were all supported. Study 1 involved an online survey with 310 participants and implies that EFL learners with greater self-efficacy tend to challenge themselves by learning content above their current proficiency. Accordingly, Study 1 provides empirical evidence for the association between EFL learners’ self-efficacy beliefs and their willingness to take on challenges to learn something above their current level of proficiency. Thus, the results of Study 1 answer the “why” question about the necessity of improving EFL learners’ self-efficacy beliefs. The question about “how” to achieve this goal of strengthening EFL learners’ self-efficacy beliefs in online learning was addressed by studies 2 and 3, which confirmed that using emojis as positive feedback can reinforce EFL learners’ self-efficacy beliefs in online settings, even when they are not attending to feedback. EFL learners’ responses to emojis may be due to the fact that people tend to use emojis to express their feelings with friends [19]. In online language learning contexts, using emojis in a positive and appropriate way can create a positive atmosphere for learning [55] and can serve as an “atmosphere-building device” in Japanese settings [82]. Since cultural differences exist in the use of emojis for communication [83], we assert that positive emojis can further act as a “self-efficacy building device” for Taiwanese EFL learners.

The theoretical implications of this research are twofold. First, it adds to the body of knowledge on the “why” and “how” of enhancing EFL learners’ self-efficacy beliefs in online contexts. By carrying out our three studies, we holistically and systematically tackled the aforementioned issues with empirical evidence. Second, we have proven that positive feedback via emojis has great potential to strengthen EFL learners’ self-efficacy beliefs, even when such feedback is subliminal. Future studies should go beyond this stream of research through the lens of neuroscience. Nurhadi and Saifullah (2021) revealed that both supraliminal and subliminal stimuli can make alpha brainwaves increase and beta brainwaves decrease [84]. Moreover, everyone’s brains deal with supraliminal and subliminal messages differently [84]. There is still a lot of unknown knowledge to discover. Furthermore, the study offers implications for educational practitioners. Positive feedback is critical for EFL learners, and therefore EFL teachers should provide positive feedback to their students [85] even when the feedback is not attentive to them. Moreover, in the context of CALL, the use of emojis for providing positive feedback is encouraged as emojis can “depict a myriad of different facial expressions with varying levels of detail” (p. 2) [86] and students are keen to use them for CMC in virtual environments [87,88].

Nevertheless, the study also had some limitations. First, H1 was rather predictable. It would be interesting to investigate it further by testing the effectiveness of self-efficacy beliefs on learning new difficult words, in addition to simply willing to learn them. Therefore, future studies can be undertaken to shed light on this issue. Second, whether positive feedbacks affect effective learning was not investigated due to the fact that they are very positive feedbacks or they are simply feedbacks. Future research can explore comparisons with more or less negative feedbacks or modulations of positive feedbacks. Third, the sample of the study was rather homogenous, as all of the participants were EFL learners from the same country/region. Malik, Qin, Oteir, and Soomro (2021) pointed out that EFL learners’ geographic background may significantly affect their anxiety pertaining to learning English, which is related to their self-efficacy beliefs [89]. Last but not the least, the three studies conducted as part of the present research were cross-sectional, which may serve as a limitation for generalizability of long-term effect of positive emoji feedback. For example, it would be interesting to examine whether the participants in Studies 2 and 3 who showed an increased level of self-efficacy in relation to positive feedback (subliminal and supraliminal) attempted the task of Study 1 (i.e., choosing which words [easy vs. difficult] they would like to learn). By examining this, whether the self-efficacy induced by positive feedbacks affects such kind of choice can also be tested. Furthermore, more comprehensive information with regard to this can be acquired through longitudinal research.

To the best of our knowledge, the present research is the first to involve three studies to address one of the most important topics pertaining to online learning, specifically, the effect of using emojis for feedback. We extracted empirical data from three research designs with three different groups of participants. Our results underscore the importance of enhancing EFL learners’ self-efficacy beliefs and how effective positive feedback via emojis can achieve this aim. Furthermore, positive feedback via emojis, presented subliminally, can be as effective as supraliminal feedback. In other words, as long as there is positive feedback provided through emojis, it will be able to reinforce EFL learners’ self-efficacy beliefs, regardless of whether students attend to the feedback.

## Figures and Tables

**Figure 1 behavsci-12-00227-f001:**
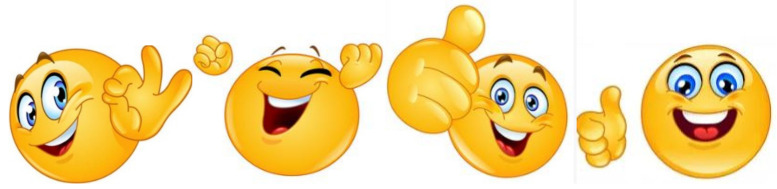
Four positive emojis used in the study (accessed on 15 December 2021 from https://www.vectorstock.com/royalty-free-vectors/vectors-by_yayayoy).

**Figure 2 behavsci-12-00227-f002:**
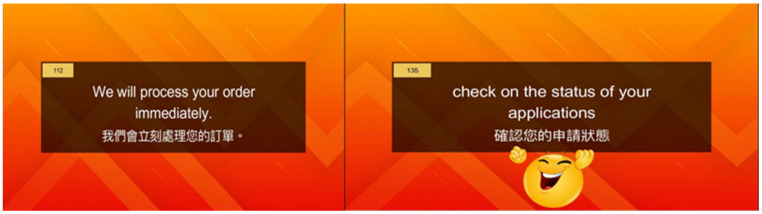
Learning materials with and without positive feedback provided through emojis. Note: The Chinese language is the translation of English above.

**Figure 3 behavsci-12-00227-f003:**
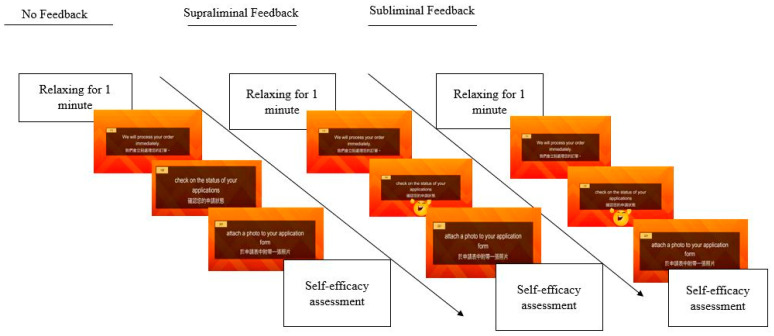
Research procedure (Study 3).

**Table 1 behavsci-12-00227-t001:** Fit statistics for single-factor CFA of the QESE reading section.

Model	CMIN/df	NFI	CFI	TLI	RMSEA	RMSEA 90% CI
Single-Factor	1.87	0.98	0.98	0.98	0.05	0.00–0.09

Note: NFI = Norm Fit Index; CFI = Comparative Fit Index; TLI = Tucker Lewis Index RMSEA = Root Mean Square Error of Approximation.

**Table 2 behavsci-12-00227-t002:** Fit statistics for single-factor CFA of the QESE listening portion.

Model	CMIN/df	NFI	CFI	TLI	RMSEA	RMSEA 90% CI
Single-Factor	1.93	0.96	0.98	0.96	0.08	0.00–0.15

Note: NFI = Norm Fit Index; CFI = Comparative Fit Index; TLI = Tucker Lewis Index RMSEA = Root Mean Square Error of Approximation.

**Table 3 behavsci-12-00227-t003:** The paired sample *t*-tests of the control and experimental groups.

	Mean	SD	*t*	*p*
CG pre-test	4.00	0.64	−0.95	0.35
CG post-test	4.07	0.38		
EG pre-test	4.05	0.50	−2.92	0.005
CG post-test	4.27	0.49		

**Table 4 behavsci-12-00227-t004:** Descriptive statistics (Study 3).

		Mean	SD	SEM
Listening	No	3.8833	0.18647	0.03404
	Supraliminal	4.0167	0.14746	0.02692
	Subliminal	4.0667	0.17287	0.03156
	Total	3.9889	0.18495	0.01950
Reading	No	3.8278	0.26437	0.04827
	Supraliminal	4.1056	0.26072	0.04760
	Subliminal	4.1778	0.23543	0.04298
	Total	4.0370	0.29326	0.03091

**Table 5 behavsci-12-00227-t005:** The results of ANOVA from Study 3.

	Source of Variance	SS	*df*	*MS*	*F*	*p*	$\eta_{p}^{2}$
Listening	Between-group	0.54	2	0.27	9.36	0.00	0.18
	Within-group	2.51	87	0.03			
	Total	3.04	89				
Reading	Between-group	2.05	2	1.02	15.90	0.00	0.27
	Within-group	5.61	87	0.06			
	Total	7.65	89				

## Data Availability

The data that support the findings of this study are available from the corresponding author upon reasonable request.

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
