# Peer review of "Enhancing EFL Learners’ Self-Efficacy Beliefs of Learning English with Emoji Feedbacks in CALL: Why and How"

_behavsci, 2022, doi:10.3390/bs12070227_

Round 1

Reviewer 1 Report

Thank you for the opportunity to review this interesting manuscript. Overall, I found the it persuasive, and well presented. I have some comments that I think need to be addressed before the manuscript can be considered for publication. The manuscript examines the positive relationship between positive feedback (in terms of emojis), presented in three conditions of consciousness, and learning (in terms of self-efficacy beliefs and willing to learn challenging words). This is clearly a significant topic and issue.

 Introduction

-The introduction states that the feedback given through emojis “is consciously and/or subconsciously perceived by the student". It is not clear which information is processed consciously and/or subconsciously. Also in the introduction, the concept of “subliminal feedback” is introduced without explaining it (page 2, line 64), immediately afterwards it moves on to “subliminal information”, which can also be perceived in an unconscious state (line 65). Thus, it is not clear what is meant by “subliminal feedback”, whether it is conscious and/or unconscious, why emojis are examples of subliminal feedback. The connections between these concepts become clearer throughout the paper. I suggest better developing and defining them from the beginning, in order to focus more clearly on the object of the research.

-On page 2, lines 49-50, it is stated that "most EFL learners do not seem to have a high self-efficacy belief about learning English (Åžener & Erol, 2017)". The statement is a very generalised one but it is not clear on what argumentative or empirical basis it rests. Indeed, a single study is hardly sufficient to make such a generalisation, unless it is a meta-analysis work. I would suggest better explaining what Åžener & Erol found to show the scope of generalisation.  Unfortunately, the reference to their work is not in the reference section, so it should be included.

-Referring to the conscious/subconscious issue, in the introduction the authors say that "the unconscious mind is more powerful than the conscious one (Naz, 2016)". In what ways the unconscious mind is more powerful than the conscious one?

-On page 3, line 106, the authors introduce the variable under consideration in study 1 in a vague and generalized way: “willing to accept challenges”. I suggest circumscribing the kind of challenge under investigation.

 Study 1

-What criteria were used to distinguish difficult words from easy ones? The procedure is unclear: were lists of words presented to the participants? How was the participants' willingness to learn one type or the other of words measured? Was a Likert scale or a categorical scale used to measure this task?

 Study 2 and Study 3

-In study 2 positive feedback lasting 3 seconds and appearing every 5 questions are provided. The same type of feedback is used in Study 3, but in study 3 this type of feedback is referred to as “supraliminal feedback”. I suggest using the same label for both of them  and specifying that the feedback always used were positive ones.

 Study 3

-Please, report the studies on the basis of which a feedback presented for 0.3 seconds can be defined as a subliminal feedback.  

-Was the order of presentation of the three types of feedback randomised? If not, this should be discussed as a limitation.

 Study 1 Study 2 and 3

-It is not completely clear whether some of the samples of participants were the same in the three studies. Please clarify.

-I guess that the feedbacks used in the three studies were always positive. I suggest making this information explicit.

 Results and discussion

-On page 6, lines 282-283, the authors say that “learners with greater self-efficacy beliefs tend to challenge themselves by learning content above their current proficiency”. This affirmation is interpretative and not driven by the results. Participants were not asked whether the new words were challenging for them, neither the levels of perceived difficulty were not measures.  

-The results of study 3 with regard to the three conditions are somewhat confusing. I suggest making explicit which condition was associated with each of the three lessons. I also suggest reporting the data of the three conditions in a table.

-The lack of difference between the supraliminal and subliminal condition (lines 332-333) should be discussed. References to Dao et al (2021) and González-Lloret (2015) do not help, because they concern the use of emoij.

- On page 8, line 336, the authors say that “EFL learners not only like to use emojis for CMC, but also prefer to receive them as feedback”. These two statements are not driven from the studies, since participants were not asked whether they liked using emojis or whether they preferred receiving them as feedback.

 General Discussion, Implications, and Conclusions

-The limitations of the research were not highlighted. One of these, for example, is that Hypothesis 1 is rather predictable. It would be interesting to investigate it further by testing the effectiveness of self-efficacy beliefs on learning new difficult words, in addition to simply willing to learn them.

-Furthermore, study 1 does not clarify whether participants with higher levels of self-efficacy perceive the new difficult words as more challenging and/or less difficult than participants with lower levels of self-efficacy.

-In order to have further proof of the result of Study 1, it would be interesting to check whether the participants in Study 2 and 3 who showed an increased level of self-efficacy in relation to positive feedback (subliminal and supraliminal) were subjected to the task of Study 1, i.e. choosing which words (easy vs. difficult) they would like to learn. In this way, it could be tested whether the self-efficacy induced by positive feedbacks affects this type of choice.

-Another limit could be that it has not been investigated whether positive feedbacks impact on effective learning due to the fact that they are very positive feedbacks (an exultant smile is reported in figure 1) or simply they are feedbacks. Comparisons with more or less negative feedbacks or modulations of positive feedbacks could be argued.

Author Response

Responses to the comments of reviewer 1

  1. The introduction states that the feedback given through emojis “is consciously and/or subconsciously perceived by the student". It is not clear which information is processed consciously and/or subconsciously. Also in the introduction, the concept of “subliminal feedback” is introduced without explaining it (page 2, line 64), immediately afterwards it moves on to “subliminal information”, which can also be perceived in an unconscious state (line 65). Thus, it is not clear what is meant by “subliminal feedback”, whether it is conscious and/or unconscious, why emojis are examples of subliminal feedback. The connections between these concepts become clearer throughout the paper. I suggest better developing and defining them from the beginning, in order to focus more clearly on the object of the research.

Response:

Thank you for the reviewer’s question, comments, and suggestions. We have revised the paper to clarify the unclear parts as follows:

 “Per suggestions of previous studies, Ionescu (2016) proposed that any stimuli that last 30-50 milliseconds can be considered as subliminal stimuli. Weber (2016) and Carr (2010) further set up the threshold value of subliminal stimuli as 30 milliseconds. In this current study, subliminal feedback referred to the feedback that appears less than .03 second (or 30 millisecond) and unconsciously attained by the receivers. When such feedback was presented with emoji and would be unconsciously perceived by the students, these were subliminal feedback in emoji. Conversely, when conscious feedback in emoji were captured consciously by the students, these would be supraliminal feedback.” (please see p.2).

  1. On page 2, lines 49-50, it is stated that "most EFL learners do not seem to have a high self-efficacy belief about learning English (Åžener & Erol, 2017)". The statement is a very generalised one but it is not clear on what argumentative or empirical basis it rests. Indeed, a single study is hardly sufficient to make such a generalisation, unless it is a meta-analysis work. I would suggest better explaining what Åžener & Erol found to show the scope of generalisation. Unfortunately, the reference to their work is not in the reference section, so it should be included.

Response:  

Many thanks for the reviewer’s comments and we totally agree with the reviewer’s viewpoint; hence, we tried to address this issue by adding more empirical work such as Ngo and Eichelberger (2021), SaÄŸlamel and AydoÄŸdu (2022) as well as Zhang (2018). Additionally, the citation of Sener and Erol (2017) was included.

Ngo, H., & Eichelberger, A. (2021). College Students' Perceived Self-Efficacy and Use of Information and Communication Technologies in EFL Learning. International Journal of Education and Development using Information and Communication Technology17(1), 34-44.

SaÄŸlamel, H., & AydoÄŸdu, Z. M. (2022). The Academic Writing Needs of Students: A Case Study on Stakeholder Perspectives. Acuity: Journal of English Language Pedagogy, Literature and Culture7(1), 37-50.

Sener, S., & Erol, I. K. (2017). Motivational orientations and self-efficacy beliefs of Turkish students towards EFL learning. Eurasian Journal of Educational Research, 67, 251-267. DOI: http://dx.doi.org/10.14689/ejer.2017.67.15.

Zhang, Y. (2018). Exploring EFL Learners' Self-Efficacy in Academic Writing Based on Process-Genre Approach. English Language Teaching11(6), 115-124.

  1. Referring to the conscious/subconscious issue, in the introduction the authors say that "the unconscious mind is more powerful than the conscious one (Naz, 2016)". In what ways the unconscious mind is more powerful than the conscious one?

Response:

We are grateful for the reviewer’s constructive suggestions, and we have added related information accordingly as follows:

 “As being put forth by Bargh and Morsella (2008) ‘the unconscious mind is a perva-sive, powerful influence over such higher mental processes’ (p. 73). Hamad (2007) further pointed out that one's unconscious mind would be able to facilitate his/her creativity and accomplish what the conscious one could not.” (please see p. 2)

Bargh, J. A., & Morsella, E. (2008). The unconscious mind. Perspectives on Psychological Science3(1), 73-79.

Hamad, S. (2007). Creativity: Method or magic?. In Consciousness and cognition (pp. 127-137). Cambridge, MA: Academic Press.

  1. On page 3, line 106, the authors introduce the variable under consideration in study 1 in a vague and generalized way: “willing to accept challenges”. I suggest circumscribing the kind of challenge under investigation.

Response:

Thanks a lot for the reviewer’s comments. Yes, we did not elaborate this part in a clearer way and we revised this part by providing more detailed information about the challenges we stated in the text.

 “challenges which in this study refers to the vocabulary lists that were beyond the participants’ current vocabulary volume or were above their current level of proficiency.”

  1. Study 1-What criteria were used to distinguish difficult words from easy ones? The procedure is unclear: were lists of words presented to the participants? How was the participants' willingness to learn one type or the other of words measured? Was a Likert scale or a categorical scale used to measure this task?

Response:

Many thanks for the reviewer’s questions. Please allow us to address them in detail and hopefully our answers can meet the reviewer’s expectations.

 “the dependent variable of study 1 was categorical. The levels of difficulty about vocabulary lists were based on the participants’ level of proficiency. Challenging words were those selected from ‘Word Smart for the GRE’ and these vocabulary lists were difficult to EFL learners (Park, Kim, Kim, & Mun, 2019) whereas easy words were those that were used in daily conversation.”

Park, J., Kim, S., Kim, A., & Mun, Y. Y. (2019). Learning to be better at the game: Performance vs. completion contingent reward for game-based learning. Computers & Education139, 1-15.

  1. Study 2 and Study 3-In study 2 positive feedbacks lasting 3 seconds and appearing every 5 questions are provided. The same type of feedback is used in Study 3, but in study 3 this type of feedback is referred to as “supraliminal feedback”. I suggest using the same label for both of them and specifying that the feedback always used were positive ones.

Response:

Many thanks for the reviewer’s suggestions which would increase the comprehensibility of this manuscript. We changed the description of the supraliminal feedback as 3-second feedback in Study 3 as they were also positive. (please see p. 6 and p. 7 for details).

  1. Study 3- Please, report the studies on the basis of which a feedback presented for 0.3 seconds can be defined as a subliminal feedback.

Response:

Thanks for the reviewer’s suggestions, Ionescu (2016) proposed that any stimuli that last 30-50 m.s. can be considered as subliminal stimuli. Weber (2016) and Carr (2010) further set up the threshold value of subliminal stimuli as 30 m.s.  See p. 2 for the revision we did on this part.

Carr, L. (2010). The influence of nonverbal behavior and empathic listening on the early development of the working alliance: Therapist reflections on first sessions. San Francisco, CA: Alliant International University.

Ionescu, M. R. (2016). Subliminal perception of complex visual stimuli. Romanian Journal of Ophthalmology60(4), 226.

Weber, D. (2016). Brand seduction: How neuroscience can help marketers build memorable brands. Newburyport, MA: Red Wheel/Weiser.

  1. Study 3-Was the order of presentation of the three types of feedback randomised? If not, this should be discussed as a limitation.

Response:

We appreciated the reviewer’s question, and the answer is yes, the order of presentation of the three types of feedback randomized. We have explicitly stated this part in the text as “We randomized the sequence of these three lessons and made the interval between each participant’s lessons at least 48 hours to prevent the carryover effect (Brooks, 2012).” (p. 7, line 283-285).

  1. It is not completely clear whether some of the samples of participants were the same in the three studies. Please clarify.

Response:

Many thanks for the reviewer’s viewpoint, samples of participants were not the same in three studies as the nature of these three studies were not identical. Accordingly, different numbers of participants were recruited based on the results of G*Power calculation.

  1. I guess that the feedbacks used in the three studies were always positive. I suggest making this information explicit.

Response:

We are grateful for the reviewer’s suggestion, and we have explicitly stated this in the text.

  1. On page 6, lines 282-283, the authors say that “learners with greater self-efficacy beliefs tend to challenge themselves by learning content above their current proficiency”. This affirmation is interpretative and not driven by the results. Participants were not asked whether the new words were challenging for them, neither the levels of perceived difficulty were not measures.

Response:

Thank you very much for the reviewer’s constructive suggestion. This sentence was indeed too interpretative; therefore, this sentence has been rewritten as “learners with greater self-efficacy tend to choose the vocabulary lists from GRE workbook whilst those who scored lower on self-efficacy were keen to select the vocabulary list that appeared more often in the daily life.”  (p. 7, Line 300-302)

  1. The results of study 3 with regard to the three conditions are somewhat confusing. I suggest making explicit which condition was associated with each of the three lessons. I also suggest reporting the data of the three conditions in a table.

Response:

Many thanks for the reviewer’s suggestions. We have revised Study 3 accordingly to provide detailed information about the experiment design. We also included descriptive data of three groups in Table 4.

  1. The lack of difference between the supraliminal and subliminal condition (lines 332-333) should be discussed. References to Dao et al (2021) and González-Lloret (2015) do not help, because they concern the use of emoij.

Response:

Thanks a lot for the reviewer’s comments and we added the research of Ham, Midden and Beute (2009) to elaborate the findings of our study. Sentence as follows was added in p. 9

“Concerning the effectiveness of subliminal and supraliminal emoji as feedback, the results of this study unveiled that both two types of emoji feedback were identical, and they could significantly increase EFL learners’ self-efficacy compared to no feedback. Such a findings were in line with Ham, Midden and Beute (2009) who discovered that effectiveness of supraliminal and subliminal feedback were similar; this current study further advances the current understanding based on their research by explicating emoji feedback can also work.”

Ham, J., Midden, C., & Beute, F. (2009, April). Can ambient persuasive technology persuade unconsciously? Using subliminal feedback to influence energy consumption ratings of household appliances. In Proceedings of the 4th International Conference on Persuasive Technology (pp. 1-6).

  1. On page 8, line 336, the authors say that “EFL learners not only like to use emojis for CMC, but also prefer to receive them as feedback”. These two statements are not driven from the studies, since participants were not asked whether they liked using emojis or whether they preferred receiving them as feedback.

Response:

Thanks a lot for the reviewer’s comments. We revised this part as “Study 3 expands the current understanding provided by the first two studies by showing the effectiveness of positive emoji feedback exploited for EFL learners’ learning in CALL settings, regardless of whether the feedback was subliminal or supraliminal.” (please see p. 9).

  1. The limitations of the research were not highlighted. One of these, for example, is that Hypothesis 1 is rather predictable. It would be interesting to investigate it further by testing the effectiveness of self-efficacy beliefs on learning new difficult words, in addition to simply willing to learn them.

Response:

We appreciated the reviewer’s advice and we added research limitation in p.10 and p.11 as follows:

“Limitations of this study are stated as follows: firstly, Hypothesis 1 is rather predictable. It would be interesting to investigate it further by testing the effectiveness of self-efficacy beliefs on learning new difficult words, in addition to simply willing to learn them. Therefore, future studies are advised to shed lights on this issue. Another limit could be that this present study did not investigate whether positive feedbacks impact on effective learning due to the fact that they are very positive feedbacks or simply they are feedbacks. Comparisons with more or less negative feedbacks or modulations of positive feedbacks could be argued and future researches are advised to explore on this issue. Thirdly, the participants of these three studies were rather homogenous as they were all EFL learners from the same country/region. As Malik, Qin, Oteir, and Soomro (2021) pointed out that EFL learners’ geographic background would significantly affect their anxiety of learning English which was related to their self-efficacy beliefs.

Last but not least, the three studies of this current research were cross-sectional which would limit the generalizability of long-term effect of positive emoji feedback. For example, it would be interesting to examine whether the participants in Study 2 and 3 who showed an increased level of self-efficacy in relation to positive feedback (subliminal and supraliminal) were subjected to the task of Study 1, i.e. choosing which words (easy vs. difficult) they would like to learn. In this way, it could be tested whether the self-efficacy induced by positive feedbacks affects this type of choice. More comprehensive information such as this can be acquired if longitudinal research can be designed in the future.”

Malik, S., Qin, H., Oteir, I., & Soomro, M. A. (2021). Detecting Perceived Barriers in FLSA: The Socio-Psycholinguistic Study of EFL University Learners. Advances in Language and Literary Studies12(1), 34-45.

  1. Study 1 does not clarify whether participants with higher levels of self-efficacy perceive the new difficult words as more challenging and/or less difficult than participants with lower levels of self-efficacy.

Response:

Thanks for the reviewer’s comments and yes, participants of Study 1 did not clarify whether participants with higher levels of self-efficacy perceive the new difficult words as more challenging and/or less difficult than participants with lower levels of self-efficacy. Therefore, results of Study 1 were rewritten as “The findings of Study 1 provide empirical support for H1, which is that EFL learners with greater self-efficacy tend to choose the vocabulary lists from GRE workbook whilst those who scored lower on self-efficacy were keen to select the vocabulary list that appeared more often in the daily life.” (p.7, Line 299-301).

  1. In order to have further proof of the result of Study 1, it would be interesting to check whether the participants in Study 2 and 3 who showed an increased level of self-efficacy in relation to positive feedback (subliminal and supraliminal) were subjected to the task of Study 1, i.e. choosing which words (easy vs. difficult) they would like to learn. In this way, it could be tested whether the self-efficacy induced by positive feedbacks affects this type of choice.

Response:

We really appreciate the reviewer’s suggestion. It is indeed a great idea to examine the effect in this way; nevertheless, the three studies have been conducted and we did not come up with such a great idea back at that time. We also did not have enough time to redo the experiments; hence, we put this suggestion asone of the research limitations as follows:

“Last but not least, the three studies of this current research were cross-sectional which would limit the generalizability of long-term effect of positive emoji feedback. For example, it would be interesting to examine whether the participants in Study 2 and 3 who showed an increased level of self-efficacy in relation to positive feedback (subliminal and supraliminal) were subjected to the task of Study 1, i.e. choosing which words (easy vs. difficult) they would like to learn. In this way, it could be tested whether the self-efficacy induced by positive feedbacks affects this type of choice. More comprehensive information such as this can be acquired if longitudinal research can be designed in the future.” (p.10 and p.11).

  1. Another limit could be that it has not been investigated whether positive feedbacks impact on effective learning due to the fact that they are very positive feedbacks (an exultant smile is reported in figure 1) or simply they are feedbacks. Comparisons with more or less negative feedbacks or modulations of positive feedbacks could be argued.

Response:

Thanks for the reviewer’s suggestions again. We include this wonderful and constructive comments as one of the major research limitations on p. 10.

Reviewer 2 Report

In the conclusions, it is necessary to incorporate the projections of the study in the field of education. It must incorporate the possible applications and scope of the study. As well as add some more updated reference

Author Response

Responses to the comments of reviewer 2

  1. In the conclusions, it is necessary to incorporate the projections of the study in the field of education. It must incorporate the possible applications and scope of the study. As well as add some more updated reference

Response:

We appreciate the reviewer’s constructive suggestion, and we add related information as well as updated information in the conclusion section as “Furthermore, implications for educational practitioners that this present research include that positive feedback are critical for EFL learners and therefore EFL teachers are suggested to provide positive feedback to their students (Xu, 2015) even when the feedback are not attentive to them. Also, in the context of CALL, the uses of emoji as positive feedback are encouraged as emoji can “depict a myriad of different facial expressions with varying levels of detail” (Holtgraves & Robinson, 2021, p. 2) and students are keen to use emoji for CMC at virtual environments (Tang, Zhao, Chen, & Zhao, 2021).” 

Helens-Hart, R., & Carlson, G. (2022). Using emoji to practice impromptu speaking. Communication Teacher36(2), 122-126.

Tang, M., Zhao, X., Chen, B., & Zhao, L. (2021). EEG theta responses induced by emoji semantic violations. Scientific Reports11(1), 1-9.

Xu, Y. (2015). The necessity of appropriate application of multimedia instruction to English teaching. Canadian Social Science11(3), 304-308.

Reviewer 3 Report

Although the idea underlying this piece of research is interesting I believe the paper needs to reconsider several issues: 

1. ABSTRACT.  Reconsider  ‘For Study 1, an online survey with 310 participants, we ascertained the extent to which EFL learners (lines 12-13). Should it be ‘based on’ an online survey’ or ‘we ascertained […] through an online survey..? 

2. HYPOTHESES and PREVIOUS WORKS. The following statement includes the only reference about the negative consequences of using emojis since the rest of cited works highlight the positive impact: ‘Al-Garaady and Mahyoob (2021) revealed that the EFL learners in their study liked to use emojis for communication, but that such an application could negatively affect their writing skills (130-131). How could it negatively affect the writing skills? The authors should provide further details and more references about the negative side, if any, of using emojis to avoid a confirmation bias in their research. 

3. HYPOTHESES. The authors should always include ‘beliefs’ after self-efficacy as this study is based on self-perceived capacity or beliefs and not on actual learning outcomes. Although they claim positive self-efficacy beliefs can be a predictor of actual performance according to different studies, such performance is not examined in this piece of research. For example RH1 (lines 111-112) reads as follows ‘RH1: EFL learners with greater self-efficacy tend to challenge themselves by learning content above their current proficiency’. To avoid confusion, this RH should be ‘EFL learners with greater self-efficacy BELIEFS tend to ..’ 

4. HYPOTHESES. RH2 and RH3 are very similar. The authors should reconsider if RH2 could be more accurately expressed stating the type of positive feedback as in ‘RH2: Positive (conscious or expl¡cit?) feedback in the form of emojis will significantly enhance EFL learners’ self-efficacy beliefs (line 141). The only different with RH3 is ‘subliminal’ feedback although section 2.3 refers to both supraliminal and subliminal but RH3 just includes subliminal, which is the only difference between RH2 and RH3. The authors should better clarify the difference between RH2 and RH3 in their formulation. 

5. HYPOTHESES. The main problem with this manuscript is the confirmation bias. RH2 and RH3 seem to be formulated to reach the desired conclusions, which is quite common in research about self-perceived skills and beliefs. A more neutral approach is recommended throughout the paper, for example: 

a)     a)    formulate the research questions in more moderate way, as in ‘RH3 Positive subliminal feedback in the form of emojis will significantly enhance EFL learners’ self-efficacy beliefs ‘ (159). The authors here are not examining whether (positive and/or negative) subliminal feedback in the form of emojis has an impact on EFL learners’ self-efficacy beliefs (in a positive or negative way)’, they seem to be inclined to ‘confirm’ positive feedback ‘enhances’ self-efficacy beliefs.

d      b)  the review of previous works and formulation of the RH seem to be specifically designed to demonstrate their ideas. For example, reconsider the following statement: ‘as well as whether using positive feedback in the form of emojis in an online environment would positively affect their self-efficacy beliefs.’ (lines 164-165), It might be better worded as ‘would affect their self-efficacy beliefs’. Repeating the word ‘positive’ several times in the same statement reinforces the idea of a confirmation bias.

        c) Similarly, ‘to understand whether positive feedback in the form of emojis, presented subliminally, would have a similar impact on boosting EFL learners’ self-efficacy beliefs  (lines 167-168). The word ‘boosting’ does not seem to be adequate when formulating a research question from a neutral perspective.

          d)  One more example, ‘For Study 1, we aimed to confirm whether EFL learners’ self-efficacy beliefs would produce a significant effect regarding their willingness to learn challenging English vocabulary.’ (lines 171-173) Confirm or reject? Measure the effect?

       In fact, the authors occasionally seem to adopt a more moderate position, as in the statement ‘For Study 2, we aimed to examine how feedback provided via emojis affects EFL learners’ self-efficacy beliefs in CALL settings ‘‘ (lines 206-207). This formulation is more neutral than the previous examples.

 6. METHOD & SAMPLING (lines 176-190). What sampling technique was used for the participants who were recruited through social media platforms?  Was it a non-probabilistic sampling technique, purposeful sampling? The authors should specify it.

7. PARTICIPANTS (lines 176-190). How did the authors know the TOEIC scores if the participants were recruited through social networks? Were those scores self-reported by the participants or did the authors have access to the actual scores? This remains unclear. 

8. METHOD & PARTICIPANTS. In the study they authors seem to have used two options: either provide positive feedback or no feedback. There is no explanation about the third option of ‘providing negative feedback’ and the impact if might have on the self-efficacy beliefs. While the authors’ approach is legitimate, they should always clarify their perspective, for example in the statement ‘EG: experimental group=CALL with feedback via emojis, appearing for 3 seconds every 5 sample questions; CG: control group=CALL without feedback via emojis’ (lines 219-221). The EG should read as ‘CALL with POSITIVE feedback via emojis’, because it seems they only option they employed.  

9. METHOD. The authors mention ‘positive feedback in the form of emojis’. Figure 1 illustrates two examples, one based on a smiley and the other one without feedback. Could the authors explain and provide further details about how many types of positive feedback in the form of emojis they used in their study? How many emojis? Which ones? 

10. METHOD. Did the participants receive the same CMC messages with (EG) and without (CG) positive feedback? Or did they get different messages? 

11. METHOD. The authors should better explain and illustrate the following statement ‘Study 3 involved a (SIC) repeated measures within-subjects (no feedback vs. supraliminal feedback vs. subliminal feedback)’ (lines 247-249). The details provided in lines 265-270 about ‘no feedback, supra and subliminal feedback’ are not clear enough. 

12. RESULTS. The authors should reconsider some statements and specify them as in ‘This outcome is in line with the work of Kim et al. (2022), who posited that students with greater self-efficacy tend to set up more challenging goals’ (lines 283-284). This should be with ‘greater self-efficacy beliefs’. 

13. RESULTS. The authors should provide some examples to strengthen their claims as in ‘Thus, EFL teachers should identify effective approaches to enhance students’ self-efficacy beliefs’ (lines 287). Provide some references. 

14. RESULTS. Could the medium effect size somehow compromise the results of Study 2? As expressed in ‘Such a significant difference is supported by a medium effect size’ (line 298) 

15. RESULTS. Some claims are not fully proven. For example, ‘the participants’ self-efficacy for reading ability was larger than their self-efficacy for listening comprehension’ (lines 320-321). How do they know? Why that difference? 

16. RESULTS. The authors need to be more specific, for example ‘To the best of our knowledge, the present research is the first to involve three studies to address one of the most important topics in online learning’ (lines 374-375). Which one? The effect of using feedback in the form of emojis? 

17. The authors should also mention some research limitations of their study.

Author Response

Responses to the comments of reviewer 3

  1. ABSTRACT. Reconsider ‘For Study 1, an online survey with 310 participants, we ascertained the extent to which EFL learners (lines 12-13). Should it be ‘based on’ an online survey’ or ‘we ascertained […] through an online survey..?

Response:

Thank you very much for the reviewer’s suggestion, we revised this part of abstract as “For Study 1, based on an online survey with 310 participants, we ascertained the extent to which EFL learners with greater self-efficacy tend to challenge themselves by learning content above their current proficiency.”

  1. HYPOTHESES and PREVIOUS WORKS. The following statement includes the only reference about the negative consequences of using emojis since the rest of cited works highlight the positive impact: ‘Al-Garaady and Mahyoob (2021) revealed that the EFL learners in their study liked to use emojis for communication, but that such an application could negatively affect their writing skills (130-131). How could it negatively affect the writing skills? The authors should provide further details and more references about the negative side, if any, of using emojis to avoid a confirmation bias in their research.

Response:

Thank you for your advice, we have added the following details based on your suggestion: “Al-Garaady and Mahyoob (2021) revealed that the EFL learners in their study liked to use emojis for communication, but that such an application might have negative effects (Lee, Kim, & Lee, 2021; Manganari, 2021); specifically, emojis could negatively affect their writing skills as their standard writing might be sacrificed because of these convenient and emergent communication tools. Liu et al. (2021) examined the use of peer feedback in face-to-face and online EFL writing courses; the Chinese EFL learners in their study indicated that emojis can be helpful in cultivating positive emotions and building rapport among peers.”

Lee, J., Kim, C., & Lee, K. C. (2021). Investigating the Negative Effects of Emojis in Facebook Sponsored Ads for Establishing Sustainable Marketing in Social Media. Sustainability13(9), 4864.

Manganari, E. E. (2021). Emoji use in computer-mediated communication. The International Technology Management Review10(1), 1-11.

  1. HYPOTHESES. The authors should always include ‘beliefs’ after self-efficacy as this study is based on self-perceived capacity or beliefs and not on actual learning outcomes. Although they claim positive self-efficacy beliefs can be a predictor of actual performance according to different studies, such performance is not examined in this piece of research. For example RH1 (lines 111-112) reads as follows ‘RH1: EFL learners with greater self-efficacy tend to challenge themselves by learning content above their current proficiency’. To avoid confusion, this RH should be ‘EFL learners with greater self-efficacy BELIEFS tend to ..’

Response:

We are grateful for the reviewer’s suggestion and the RH1 has been revised accordingly.

  1. HYPOTHESES. RH2 and RH3 are very similar. The authors should reconsider if RH2 could be more accurately expressed stating the type of positive feedback as in ‘RH2: Positive (conscious or expl¡cit?) feedback in the form of emojis will significantly enhance EFL learners’ self-efficacy beliefs (line 141). The only different with RH3 is ‘subliminal’ feedback although section 2.3 refers to both supraliminal and subliminal but RH3 just includes subliminal, which is the only difference between RH2 and RH3. The authors should better clarify the difference between RH2 and RH3 in their formulation.

Response:

Many thanks for the reviewer’s suggestions and we totally agree with this statement. Therefore we changed RH2 into “Positive feedback presented supraliminally/consciously in the form of emojis will significantly enhance EFL learners’ self-efficacy beliefs.”to clearly differentiate it from RH3 .

  1. HYPOTHESES. The main problem with this manuscript is the confirmation bias. RH2 and RH3 seem to be formulated to reach the desired conclusions, which is quite common in research about self-perceived skills and beliefs. A more neutral approach is recommended throughout the paper, for example:
  2. a)  formulate the research questions in more moderate way, as in ‘RH3 Positive subliminal feedback in the form of emojis will significantly enhance EFL learners’ self-efficacy beliefs ‘ (159). The authors here are not examining whether (positive and/or negative) subliminal feedback in the form of emojis has an impact on EFL learners’ self-efficacy beliefs (in a positive or negative way)’, they seem to be inclined to ‘confirm’ positive feedback ‘enhances’ self-efficacy beliefs.
  3. b) the review of previous works and formulation of the RH seem to be specifically designed to demonstrate their ideas. For example, reconsider the following statement: ‘as well as whether using positive feedback in the form of emojis in an online environment would positively affect their self-efficacy beliefs.’ (lines 164-165), It might be better worded as ‘would affect their self-efficacy beliefs’. Repeating the word ‘positive’ several times in the same statement reinforces the idea of a confirmation bias.
  4. c) Similarly, ‘to understand whether positive feedback in the form of emojis, presented subliminally, would have a similar impact on boosting EFL learners’ self-efficacy beliefs (lines 167-168). The word ‘boosting’ does not seem to be adequate when formulating a research question from a neutral perspective.
  5. d) One more example, ‘For Study 1, we aimed to confirm whether EFL learners’ self-efficacy beliefs would produce a significant effect regarding their willingness to learn challenging English vocabulary.’ (lines 171-173) Confirm or reject? Measure the effect?

In fact, the authors occasionally seem to adopt a more moderate position, as in the statement ‘For Study 2, we aimed to examine how feedback provided via emojis affects EFL learners’ self-efficacy beliefs in CALL settings ‘‘ (lines 206-207). This formulation is more neutral than the previous examples.

Response:

Thank you very much for the reviewer’s constructive suggestions/comments. We totally agree with these suggestions and comments and therefore have revised our manuscript accordingly. After making these changes, we also feel that we are standing on a more neutral ground to look at the proposed research questions. Below are the details:

  1. a) The third research hypothesis has been revised as “Positive feedback in the form of emojis presented subliminally and supraliminally will significantly affect EFL learners’ self-efficacy beliefs” to avoid Texas sharpshooter fallacy. (p. 4 Line 173-174).
  2. b) This sentence has been changed into “For the first two studies, we focused on the relationship between EFL learners’ self-efficacy beliefs and their willingness to challenge themselves while learning, as well as whether using positive feedback in the form of emojis in an online environment would affect their self-efficacy beliefs” (p. 4 Line 176-179) to be in compliance with the reviewer’s suggestion.
  3. c) The wording ‘boost’ has been replaced with ‘affect’ to formulate the research question from a neutral perspective. Accordingly, the sentence has been rephrased as “In terms of Study 3, we designed and administered an experiment to understand whether positive feedback in the form of emojis, presented subliminally, would have a similar impact on improving EFL learners’ self-efficacy beliefs.” (p. 4 Line 180-183).
  4. d) We have rewritten this sentence based on the reviewer’s suggestion as “For Study 1, we aimed to measure whether EFL learners’ self-efficacy beliefs would produce a significant effect regarding their willingness to learn challenging English vocabulary.” (p. 4, Line 186-188).

  1. METHOD & SAMPLING (lines 176-190). What sampling technique was used for the participants who were recruited through social media platforms? Was it a non-probabilistic sampling technique, purposeful sampling? The authors should specify it.

Response:

Thanks for the reviewer’s reminder. We did employ purposeful sampling; hence, the sampling parts of three studies have been rewritten as follows: “We administered Study 1 online and a purposeful sampling technique was adopted to recruit participants through social media platforms such as Facebook and Line groups. To ensure the representativeness of the participants…”, “Given this, a purposeful sampling was administered, and we recruited 120 EFL learners from universities in southern Taiwan; their TOEIC score was around 560 as well.” “We recruited 35 participants purposefully for Study 3 (n=35).”

  1. PARTICIPANTS (lines 176-190). How did the authors know the TOEIC scores if the participants were recruited through social networks? Were those scores self-reported by the participants or did the authors have access to the actual scores? This remains unclear.

Response:

Thanks for the reviewer’s questions. We have added two sentences to explain this part in greater details. “One open question of online survey was to ask each participant’s TOEIC score. If an individual participant who did not have taken the TOEIC or whose TOEIC was far below 560 would be removed from further analysis.” (Please see p. 5, Line 194-197).

  1. METHOD & PARTICIPANTS. In the study they authors seem to have used two options: either provide positive feedback or no feedback. There is no explanation about the third option of ‘providing negative feedback’ and the impact if might have on the self-efficacy beliefs. While the authors’ approach is legitimate, they should always clarify their perspective, for example in the statement ‘EG: experimental group=CALL with feedback via emojis, appearing for 3 seconds every 5 sample questions; CG: control group=CALL without feedback via emojis’ (lines 219-221). The EG should read as ‘CALL with POSITIVE feedback via emojis’, because it seems they only option they employed.

Response:

We truly appreciate the reviewer’s suggestion, and the description has been changed into what the reviewer suggested. (p. 6).

  1. METHOD. The authors mention ‘positive feedback in the form of emojis’. Figure 1 illustrates two examples, one based on a smiley and the other one without feedback. Could the authors explain and provide further details about how many types of positive feedback in the form of emojis they used in their study? How many emojis? Which ones?

Response:

Many thanks for the reviewer’s comments. We adopted four different positive emojis as positive feedbacks contained in online English lessons. These four emojis are shown as Figure 3 on p.6 in the manuscript. The emojis were retrieved from https://www.vectorstock.com/royalty-free-vectors/vectors-by_yayayoy.

  1. METHOD. Did the participants receive the same CMC messages with (EG) and without (CG) positive feedback? Or did they get different messages?

Response:

We are grateful for the reviewer’s question which did help to improve the comprehensibility of this manuscript. Per the questions, the answer that we provided in the text was “Given the fact that the participants received the same CMC messages with (EG) and without (CG) positive feedback We randomized the sequence of these three lessons and made the interval between each participant’s lessons at least 48 hours to prevent the carryover effect (Brooks, 2012).” (p. 7).

  1. 11. METHOD. The authors should better explain and illustrate the following statement ‘Study 3 involved a (SIC) repeated measures within-subjects (no feedback vs. supraliminal feedback vs. subliminal feedback)’ (lines 247-249). The details provided in lines 265-270 about ‘no feedback, supra and subliminal feedback’ are not clear enough.

Response:

Thanks so much for the reviewer’s suggestion. We have added more information about the design of research on p. 7 Line 273-275. Moreover, on the same page Line 289-298, detailed description of research procedure as well as a figure (Figure 3) have been included.

  1. 12. RESULTS. The authors should reconsider some statements and specify them as in ‘This outcome is in line with the work of Kim et al. (2022), who posited that students with greater self-efficacy tend to set up more challenging goals’ (lines 283-284). This should be with ‘greater self-efficacy beliefs’.

Response:

The reviewer’s suggestion is much appreciated. The sentence was rephrased as “students with greater self-efficacy beliefs tend to set up more challenging goals by taking on tasks above their present level.” (p.7).

  1. 13. The authors should provide some examples to strengthen their claims as in ‘Thus, EFL teachers should identify effective approaches to enhance students’ self-efficacy beliefs’ (lines 287). Provide some references.

Response:

Thanks a lot for the reviewer’s comment. We provide some references to strengthen our claim. The following sentences have been added “Liu, Du, Zhou, and Huang (2021) revealed that EFL learners’ self-efficacy beliefs could be enhanced by emotional arousal, particularly positive emotion (Usher & Pajares, 2008). Johnson (2021) further pointed out that direct instruction can effectively improve bilingual learners’ self-efficacy beliefs. Putting together, direct instruction along with positive feedback shown to EFL learners should be effectively boosting their self-efficacy belief.” (p.7 and p.8).

Johnson, L. R. (2021). Increasing self-efficacy of middle school emergent bilingual students. Journal of Higher Education Theory and Practice21(13), 104-113.

Usher, E. L., & Pajares, F. (2008). Sources of self-efficacy in school: Critical review of the literature and future directions. Review of Educational Research78(4), 751-796.

  1. 14. Could the medium effect size somehow compromise the results of Study 2? As expressed in ‘Such a significant difference is supported by a medium effect size’ (line 298)

Response:

Thanks for the reviewer’s question. According to Cohen (1988), a medium effect size should “represent an effect likely to be visible to the naked eye of a careful observer…”and even a small effect size is “to be noticeably smaller than medium but not so small as to be trivial…” (p. 25). (please see p.8 for details)

Cohen, J. (1988). Statistical Power Analysis for the Behavioral Sciences, 2nd Edn. New York, NY: Academic Press.

  1. 15. Some claims are not fully proven. For example, ‘the participants’ self-efficacy for reading ability was larger than their self-efficacy for listening comprehension’ (lines 320-321). How do they know? Why that difference?

Response:

It is much appreciated for the reviewer’s comment. It was indeed unclear to make such claims without related descriptive data. We have added a table to report the descriptive statistics and this section has been revised as “According to the data reported in Table 4, the participants’ self-efficacy beliefs for reading ability were higher than their self-efficacy beliefs for listening comprehension and the results of ANOVA are depicted in Table 5.” (p. 9). Hopefully, this part can be comprehended clearer in its current form.

  1. 16. The authors need to be more specific, for example ‘To the best of our knowledge, the present research is the first to involve three studies to address one of the most important topics in online learning’ (lines 374-375). Which one? The effect of using feedback in the form of emojis?

Response:

Many thanks for the reviewer’s suggestion and we have included this valuable suggestion in the text as “To the best of our knowledge, the present research is the first to involve three studies to address one of the most important topics in online learning, specifically, the effect of using feedback in the form of emojis.” (p.11).

  1. 17. The authors should also mention some research limitations of their study.

Response:

Thank you very much for the reviewer’s suggestion. We acknowledged our ignorance of not reporting limitations of this research; hence, a paragraph on limitations has been added on p.10 and p.11 as follows: “Limitations of this study are stated as follows: firstly, Hypothesis 1 is rather predictable. It would be interesting to investigate it further by testing the effectiveness of self-efficacy beliefs on learning new difficult words, in addition to simply willing to learn them. Therefore, future studies are advised to shed lights on this issue. Another limit could be that this present study did not investigate whether positive feedbacks impact on effective learning due to the fact that they are very positive feedbacks or simply they are feedbacks. Comparisons with more or less negative feedbacks or modulations of positive feedbacks could be argued and future researches are advised to explore on this issue. Thirdly, the participants of these three studies were rather homogenous as they were all EFL learners from the same country/region. As Malik, Qin, Oteir, and Soomro (2021) pointed out that EFL learners’ geographic background would significantly affect their anxiety of learning English which was related to their self-efficacy beliefs.”

Round 2

Reviewer 1 Report

I thank the authors for having addressed properly my comments and for having taken into account my suggestions.

Reviewer 3 Report

Dear authors,

Thanks for taking into consideration my comments in the revised version.